# Model tests of a 10 MW semi-submersible floating wind turbine under waves and wind using hybrid method to integrate the rotor thrust and moments

Felipe Vittori[1], José Azcona[1], Irene Eguinoa[1], Oscar Pires[1], Alberto Rodríguez[2], Álex Morató[2], Carlos Garrido[2], and Cian Desmond[3]

[1]National Renewable Energy Centre (CENER), Dept. Wind turbine analysis and design, ciudad de la innovación, 7, Sarriguren (Navarra), 31621, Spain
[2]Saitec Offshore Technologies, Parque Empresarial Ibarrabarri, Edf. A2, 48940, Leioa-Bizkaia, Spain
[3]University College of Cork, Dep. Environmental Research Institute-MAREI, Haulbowline Road, Ringaskiddy, P43C573, Cork, Ireland

**Correspondence:** José Azcona (jazcona@cener.com)

**Abstract.** This paper describes the results of a wave tank test campaign of a 1/49 scaled SATH 10 MW INNWIND floating platform. The Software-in-the-Loop (SiL) hybrid method was used to include the wind turbine thrust and the in-plane rotor moments. Experimental results are compared with a numerical model developed in OpenFAST of the floating wind turbine. The tank test campaign was carried out in the scaled model tested at the Deep Ocean Basin from the Lir National Ocean

TF at Cork, Ireland. This floating substructure design was adapted by Saitec to support the 10 MW INNWIND wind turbine within the ARCWIND project with the aim of withstanding the environmental conditions of the European Atlantic Area region. CENER provided the wind turbine controller specially designed for the SATH 10 MW configuration.

A description of the experimental set up, force actuator configuration and the numeric aerodynamic parameters are provided in this work. The most relevant experimental results under wind and wave loading are shown in time series and frequency domain.

The influence of the submerged geometry variations in the pitch natural frequency is discussed. The paper shows the simulation of a case with rated wind speed, where the tilted geometry for the computation of the hydrostatic and hydrodynamic properties of the submerged substructure is considered. This case provides a better agreement of the pitch natural frequency with the experiments, than a equivalent simulation using the undisplaced geometry mesh for the computation of the hydrodynamic and hydrostatic properties.

## 1  Introduction

Floating wind energy has experienced a great technological development with the installation of the first floating wind farms.

A relevant contribution to this technological development is the ARCWIND project (Adaptation and Implementation of Floating Wind Energy Conversion Technology for the Atlantic Region), which is a European Union funded project that aims to foster renewable energies and energy efficiency. The general objective is to reduce the technical and economic uncertainties

of floating wind technology to accelerate the up-scaling of the power capacity, making the large-scale floating projects more

commercially attractive.

With the intention to cover the most common floater topology (DNV ST 0119 , 2021) in the analysis, a tension leg platform (TLP), a spar and the SATH platform were studied under different environmental conditions of the European Atlantic region. The study presented in this document is about the SATH floating platform, a single point mooring (SPM) floater developed by Saitec (SATH , 2022). The main characteristic of this platform design is to take advantage of the weather-vanning effect to reduce the loads on its components. The floater design was up-scaled to support the rotor-nacelle-assembly (RNA) of the wind turbine 10 MW INNWIND (Bak et al., 2013). The INNWIND tower was replaced by a design from Saitec. To study the technical feasibility of this floating wind turbine concept, a scaled tank test campaign was performed in the Lir National Ocean Test Facility at the University College Cork in Ireland.

Tank testing has been an important tool for the design of offshore floating structure (Chakrabarti, 2005), (Faltinsen, 1990), (Journée and Massie, 2001). In the case of innovative floating wind turbines, it is a critical step of the design to validate the platform dynamic response subjected to the complex interactions between the wind and wave loading. Moreover, tank testing allows validating and calibrating the hydro-aero-servo-elastic numerical tools that are used for the simulation and for the loads calculations used in the components structural design and certification of the system.

To achieve a reliable reproduction of the dynamics of the full scale floating offshore wind turbine (FOWT) in the basin, it is important to obtain an accurate scaling of the relevant forces acting on the system, the inertias and the frequencies of the time variant loads. The integration of the rotor dynamics in scaled tests that combine wind and wave loading is challenging due to the scaling conflict between the Reynolds and Froude numbers that govern the aerodynamic and hydrodynamic forces (Bredmose et al., 2012) (Azcona et al., 2014b).

A hybrid testing approach named Software-in-the-Loop (SiL) was proposed and successfully applied in a test campaign by Azcona et al. (2014). In this method, the aerodynamic rotor thrust of the wind turbine is applied to the scaled model by a ducted fan or a set of propellers. The turbine thrust force is based on real-time simulations at full scale of the rotor aerody-namics, coupled with the scaled floater response that is physically tested under wave loading. The method allows considering the correctly scaled rotor load in the wave tank tests. Moreover, as the rotor loading is coupled in real time with the floater motion, the aerodynamic damping introduced by the rotor is captured. This effect is a relevant source of damping and cannot be neglected in order to accurately capture the global motions of the floating turbine. Similar methods have been applied more recently, for example by Bachynski, Chavaud, and Sauder (2015) and Belloli et al. (2020). Also, there are different approaches to introduce an aerodynamic thrust representing the full scale rotor force, such as using a drag disk Roddier et al. (2010) or building a Froude scaled rotor Koch et al. (2016).

The first version of the SiL method, where just the rotor thrust force is introduced, was successfully applied in several test campaigns for floating wind turbines. For example, in Vittori et al. (2018) the experimental measurements using this first version of SiL were compared with results from numerical computations showing good agreement and in Azcona et al. (2019) the method showed its capability to capture the low frequency dynamics of a semi-submersible. Afterwards, the SiL method was expanded to also include rotor aerodynamic and gyroscopic moments for the pitch ($M_y$) and the yaw ($M_z$) platform degrees of

freedom (DoF). This improved SiL method was used in Fontanella et al. (2020) and Vittori et al. (2020). The later, showed that the SiL method including $M_z$ is able to induce the yaw motion in the platform response. This study was done under co-linear wind and waves conditions.

   The objective of this study is to validate to improvement of the SiL method including the rotor thrust with $M_y$ and $M_z$

moments and compare the induced yaw response with numeric simulations. The measurements from experiments were used to improve the numerical model to obtain better representations of the natural frequencies. The numeric tool used was OpenFAST (NREL, 2019).

   The first section of this works gives an overview about SiL methodology applied in this test campaign, the scaled model of

the SATH10MW and the campaign setup. The second section presents a description of the OpenFAST numerical model for the SATH10MW floating wind turbine. Finally, the analysis of the results is presented in the third section, ending with the conclusions of this work.

## 2 Description of the Software-in-the-loop methodology (SiL)

The SiL hybrid method consists of replacing the rotor by a force actuator (a fan or a multipropeller system) driven by an electric motor. The scaled thrust is controlled by an electronic controller (EC) that regulates the rotational speed of the propellers motor of the actuator. This EC receives the thrust demand from a real time full scale simulation of the wind turbine. The simulation takes into account the wind field, the wind turbine control and the real time platform motions measured in the wave tank. Therefore, the method captures the coupling between the rotor loads and the platform motions, which is a relevant effect to

accurately represent the dynamics of a floating wind turbine. The FAST code developed by NREL (Jonkman , 2007) is used for the simulation of the rotor thrust loads. The details on the SiL system architecture can be in found in Azcona et al. (2014). In this test campaign, an actuator with 4 propellers was used to introduce the rotor loading. A photograph of this actuator in the calibration workbench is shown in Fig. 1.

Each of the propellers is powered by a drone commercial brushless motor that is controlled by an Electronic Speed Controller (ESC), and fed with an industrial AC/DC power supply. This system configuration produces an approximate force range of 0-24 N. The rotational speed of each motor (and therefore the force produced by the propeller) is controlled by a Pulse Width

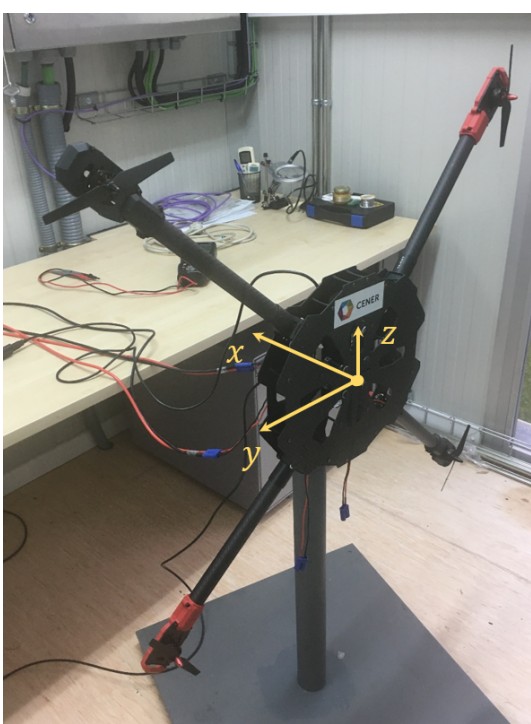

**Figure 1.** Multi-propeller actuator at CENER calibration workbench. Wind turbine thrust is applied in the $x$ direction, rotor moment $M_y$ and $M_z$ is applied in the $y$ and $z$ axis, respectively.

Modulation (PWM) signal that is generated with the LabVIEW control software, using servo libraries for Arduino. Figure 2 shows a diagram of the SiL system control scheme.

The measured motions from the tracking system of the wave tank are acquired by the SiL control scripts in LabVIEW, and

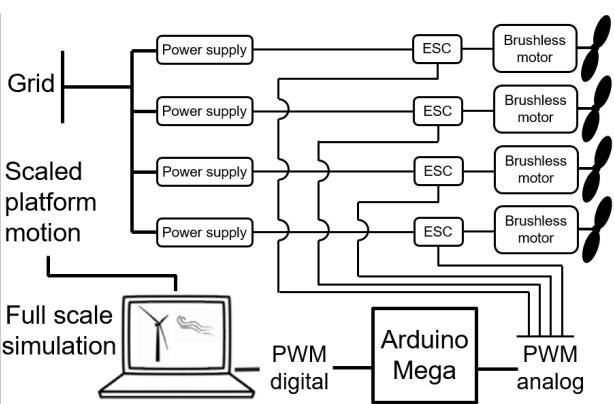

**Figure 2.** SiL control diagram. (Vittori et al., 2020)


then are integrated the simulation software for the computation of the rotor loads. This demanded rotor loading is transformed into the different propeller signals through force and moment balance equations. The propellers can work introducing only the thrust force of the rotor (each of the propellers introduces 1/4 of the scaled thrust), or the system can decouple the force that each propeller introduces, to generate the required pitch ($M_y$) and yaw ($M_z$) moments, together with the thrust. This enables the system to reproduce the scaled rotor moments from aerodynamic effects such as imbalance, wind shear, pitch failures, wind misalignment and gyroscopic effects. In this test campaign, the moments $M_y$ and $M_z$ were included in the test. The details about the development of this multi-propeller actuator can be found in Pires et al. (2020).

## 2.1 Rotor loading hybrid numerical model

A numerical model of the 10 MW INNWIND rotor was built at full scale using the FAST code coupled with AeroDyn 12.58 (Jonkman , 2007). For the execution of the experiments, it was used a modified version of this software Azcona et al. (2014) able run in real time and to integrate the measured platform motions in the computation of the rotor loads. The aerodynamic loads are based in Blade Element Momentum (BEM), tip and hub losses were considered using the Prandtl correction. The tower for the experimental scaled model in Fig. 3 was designed rigid, with a larger diameter to avoid any elastic response. The numerical model used in the hybrid testing is fed by the motions measured at the intersection between the tower center line and the water plane. For consistency with the scaled model, this numerical model assumed a rigid tower. The blades of the numerical model are also assumed rigid to improve the CPU speed ensuring real time, and because the loss of accuracy is low compared to other sources of uncertainty in the experimental setup. The turbulent wind was obtained through a Kaimal spectrum using the TurbSim wind generator from NREL (Jonkman , 2016).

The turbine controller used for the experiments was developed by CENER on the basis of state-of-the-art control strategies for pitch-controlled variable-speed turbines. Collective pitch-to-feather was applied through gain-scheduled PID controller in the above rated region, where a constant power strategy was implemented. The controller was implemented in an in-house code and compiled as a dynamic-linked library (dll) for its integration into the simulations. As for any wind turbine controller design, the controller parameters were tuned to adapt to the specific turbine dynamics of the SATH10MW platform. For such purpose, linear models were obtained from the non-linear FAST model for the whole operating wind speed range [4, 25] m/s, and an iterative design process was applied. Verification of the design was performed through non-linear simulations.

## 2.2 SATH 10MW scaled tank testing

The 1/49 SATH10MW scaled floating platform is shown in Fig. 3. This floater concept has a low draft catamaran type design with twin hull that provide stability in combination with the mooring system (SATH , 2022). In this test campaign the SPM system was not implemented to ease the uncertainties on the initial validation of the numerical tools and floating design. The impact of a SPM on the platform dynamics is expected to be relevant due to the Mz generated by the wind turbine that

will induce rotation on the platform. Therefore, is recommended further experimental campaigns to analyze this advanced
125  component. A retention system based on 4 horizontal lines separated by 90 deg between them was used to moor the system as
is shown in Fig. 3. This retention system introduce constrain in the platform yaw response avoiding the free yaw response with
respect the original SPM design.

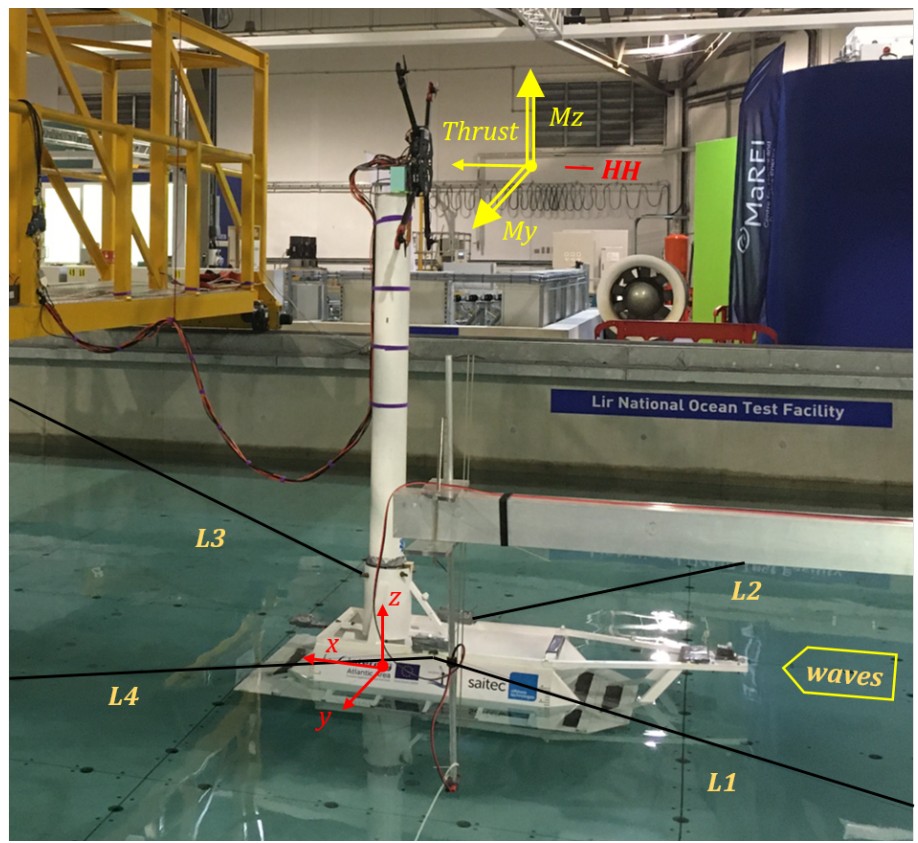

**Figure 3.** SATH10MW scaled model.

The drone frame with the four propellers that are used to introduce the scaled 10 MW INNWIND rotor loads was installed
at the tower top of the scaled floating platform. The reference system for the loads is indicated at the tower top in Fig. 3. The
130  water depth at full scale is 110 m and the wave generator produces waves in the direction also indicated in Figure 3.
The results presented in this work are based on a coordinate system located at the intersection of the MSL (Mean Sea Level)
and the tower axis indicated in Fig. 3 in red color. The geometrical center of the drone frame was located at the equivalent full
scale hub height of the wind turbine.
The resulting mass of the set of propellers together with the carbon fiber arms and the frame was relatively low, and thus ballast
135  was added in order to match the target weight that represents the full scale 10 MW INNWIND RNA mass. The COG location
and the moments of inertia were calculated based on numerical mass distribution calculations. Significant amounts of lead were

added to the heave plate, the transition piece and the nacelle to achieve the correct distribution. The difference in the moments of inertia is stimated below 1% and the CoG position below 5mm in any of the 3 directions.

## 3   SATH10MW OpenFAST numerical model for comparison

A numerical model of the SATH10MW was built at full scale in OpenFAST v2.2 (NREL, 2019) with the objective of reproducing numerically the experimental cases and compare the results with the experimental measurements. The platform added mass, damping, hydrostatic stiffness and wave force coefficients were obtained from the potential theory WAMIT (Lee and Newman , 2006) code in frequency domain and then, given as an input to OpenFAST. For the simulation of the experimental cases, the measured wave elevation time series from the tests were used as an input to OpenFAST, which generates the wave kinematics with first order wave theory.

The second order hydrodynamic forces was implemented by means of Newman approximation. Saitec provided inputs files to be used in OpenFAST. Additional linear and quadratic damping coefficients were incorporated in all platform DoF after the model damping was calibrated based on the experimental decay tests, as it is shown in Section 4.1.

The retention system of the scaled floater was modeled using a linear stiffness matrix, considering the couplings between the corresponding DoF. This linear stiffness matrix was initially dstimated analytically, based on the mooring lines tension, fairlead positions and the scaled model geometry. Afterwards, the coefficients of the matrix were tuned to match the experimental results from free decays.

The aerodynamic loads were calculated with AeroDyn v15, through the Blade Element Momentum (BEM. The tip and hub losses were considered using the Prandtl correction. This model was defined with tower and blades rigid to match the experiment conditions. The input wind field used in the simulations were the same used during the experiments to maintain consistency. Finally, the same wind turbine controller used during the tank testing was used in the numerical model.

## 4   Results discussions

This section presents some of the most relevant experimental measurement together with the numeric results obtained from the calibrated numerical model simulated by OpenFAST. The next load cases presented start with the more simple tests like free decay until the validation of more complex load scenarios like simultaneous turbulent wind and irregular waves. This allows isolating the different effects to simplify the analysis as it was also recommended by Robertson et al. (2013).

First, it is shown the results from the free decay tests, that were used to calibrate the numerical model. These results show that the platform natural frequencies and damping levels are similar to the full scale numeric model in OpenFAST.

Second, it is presented the platform response under constant and uniform wind using SiL method to introduce the rotor loads. Through these tests it was verified that the rotor loads produced a similar displacement in the experiment than the full scale

numeric model.

Finally, it is presented the similarity obtained between numerical results and measurements when the platform is under turbulent
wind only. Additionally, the effect of the new SiL is shown when the floating platform is under wave and wind loading. All
results presented in this document are at full scale.

## 4.1  Free decay tests

The free decay experimental results allowed calibrating the hydrodynamic damping levels of the OpenFAST SATH10MW
model adjusting the linear and quadratic damping terms. Also, the natural periods of the platform DoF were obtained by ad-
justing the coefficients of the stiffness matrix for the mooring system. In these experimental tests the SiL actuator was turned
off. Figure 4 (a) and (b) show the good agreement found by the OpenFAST model for surge and sway, respectively. Figure 5
(a) and (b) shows the good approximation obtained for pitch and yaw respectively.

The values of the natural periods are not shown due to confidential restrictions. The surge and sway results from Fig. 4 (a)
and (a) are referred to the center of mass of the floating wind turbine to make easier the interpretation of the DoF response.
There is strong coupling between certain DoF, such as sway and yaw when using the coordinate system from Fig. 3.

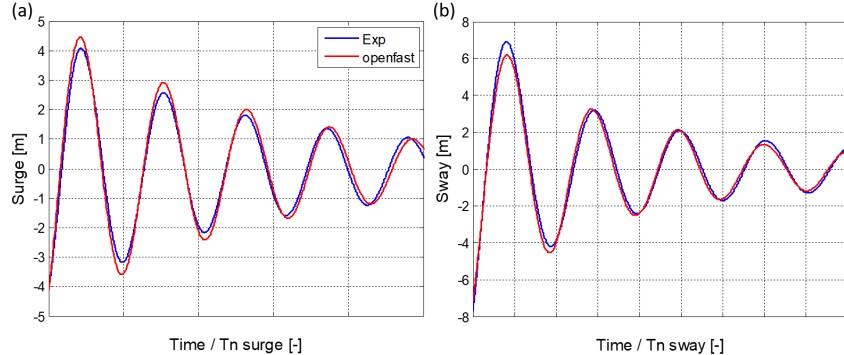

**Figure 4.** Free decay results for (a) surge and (b) sway

From experimental pitch free decays with and without cable bundle it was estimated a 3% of difference in platform pitch
stiffness. The numerical model take into account this effect by including an additional pitch stiffness coefficient. Additionally,
a pre-load moment was required to match the FOWT pitch mean position.

A good agreement is also obtained between numerical model and experiment in the yaw DoF for the firsts oscillations. Af-
terwards, yaw oscillations presents changes in natural period. This may be related with uncertainties in the estimation of yaw
stiffness in the seakeeping system that cannot be modelled numerically by linear stiffness matrix. Obviously, a SPM system

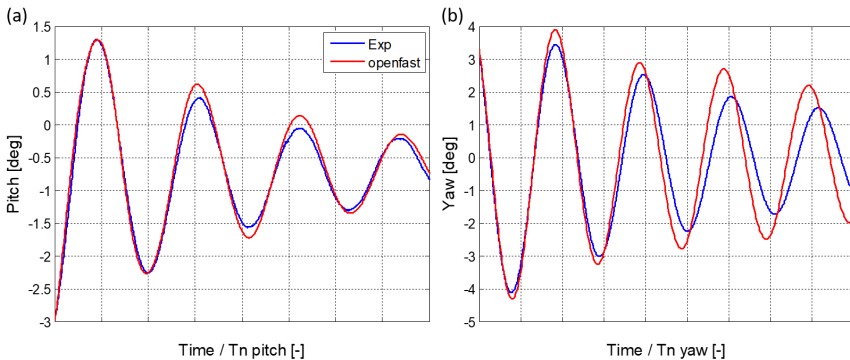

**Figure 5.** Free decay results for (a) pitch and (b) yaw

would not produce any restriction in yaw allowing to freely rotate.

For the following sections, the results will be referred to the coordinated system indicated in Fig. 3.

### 4.2   Constant and uniform winds only

Figure 6 presents the steady state response comparison for the platform surge and pitch displacements between experimental
measurements and numerical results. There are no wave and the wind is constant and uniform. The openFAST results were
very close to the experiments for 7.5 m/s case the differences between them were below 1% for surge and pitch. In the case of
11.4 m/s of wind speed the numerical result for surge was 5% larger than the experiment. The simulation solution for pitch was
2% below the tank test at turbine rated wind speed of 11.4 m/s.
The good agreement between numerical results and experimental measurements for the surge and pitch responses under both
wind speeds indicates the equivalence between the scaled experimental model and the numerical model .

### 4.3   Turbulent winds only

In this section, the motions of the platform under a turbulent wind loading, with no waves, are discussed. Figure 7 shows the
comparison between the measurements in the experiment and the computations from the equivalent simulations in OpenFAST
for the platform surge and pitch motions, under a 7.5 m/s turbulent mean wind speed. Results are presented in time domain and
in frequency domain, with a Power Spectral Density (PSD).

Time domain surge response from OpenFAST in Fig. 7 (a) matches well the response measured in the experiment. The surge
Power Spectrum Density (PSD) Fig. 7 (b) also shows the agreement between numerical and experiment results in the lower
frequencies region and around the surge natural frequency, indicated in the plot.


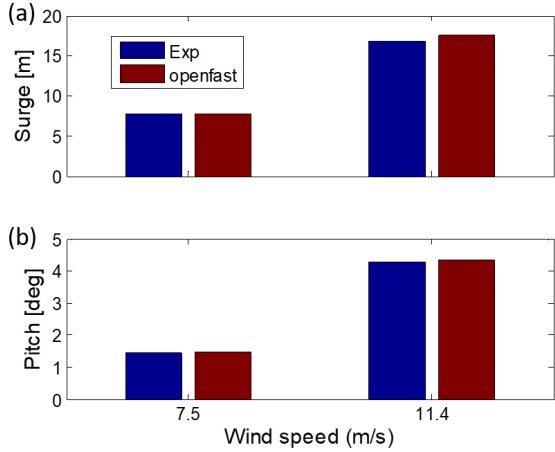

**Figure 6.** Steady state response comparison between experiments (improved SiL) and OpenFAST simulation for (a) Surge and (b) Pitch response

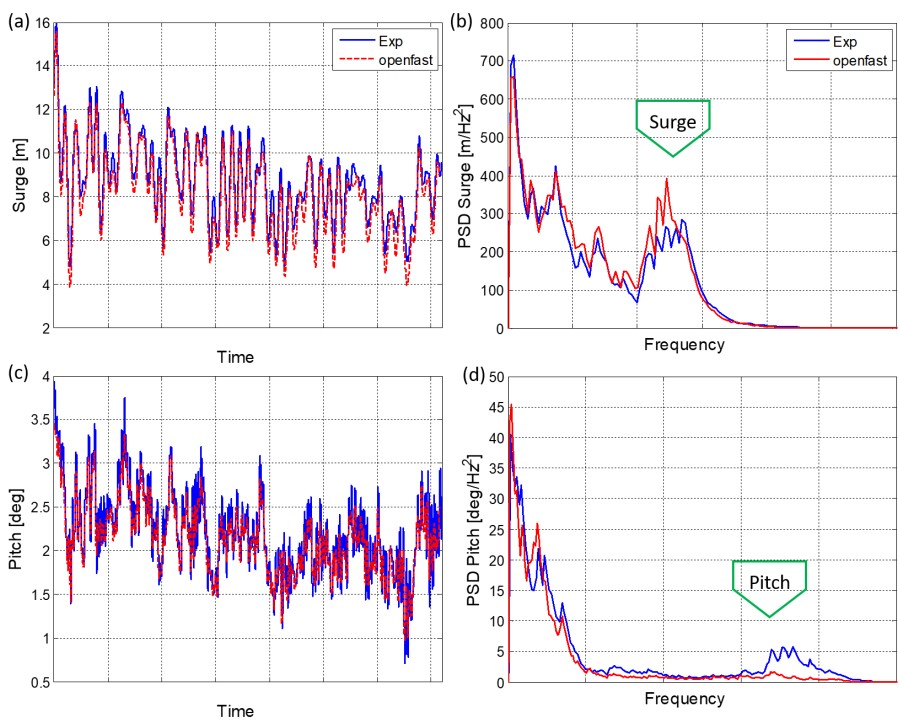

**Figure 7.** Platform response comparison between experiments and OpenFAST simulation for surge and pitch under 7.5 m/s turbulent mean wind speed without waves. (a) Surge response in time domain (b) Surge response PSD (c) Pitch response in time domain and (d) Pitch response PSD

In the case of the SATH10MW pitch response in Fig. 7 (c) the simulation results and measurements agree well for the the lower frequencies, were the wind energy is located, but for the higher frequencies, around platform pitch natural frequency, the motion is underestimated by the simulations.

Figure 8 (a) and (b) shows that there is significant difference between the measured and calculated sway response. The sway response in the experiments has larger excursions than the calculated in the numeric simulations. Additionally, the sway natural period of the scaled model seems to be slightly shifted with respect to the numerical model.

The yaw response comparison in Fig. 8 (c) shows a certain agreement between the measured scaled motion and the simulation results, although the OpenFAST solution presents lower peaks for the yaw rotations. This can be also observed in the yaw PSD
Fig. 8 (d) where the simulation curve presents lower values than the experimental.

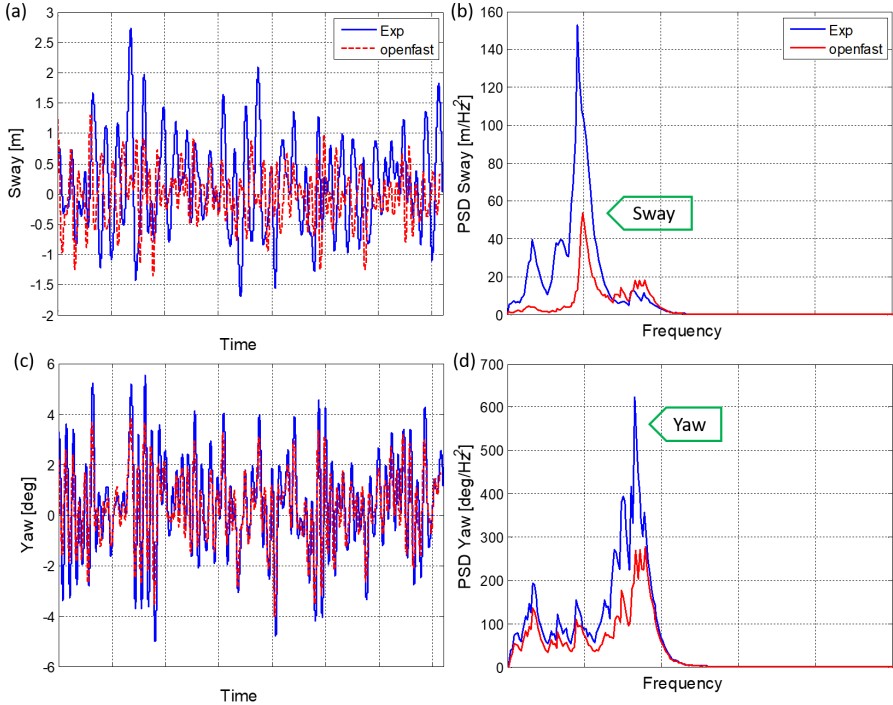

**Figure 8.** Platform response comparison between experiments and computations of sway and yaw under 7.5 m/s turbulent mean wind speed without waves. (a) Sway response in time domain (b) Sway response PSD (c) Yaw response in time domain and (d) Yaw response PSD

## 4.4    Turbulent winds and irregular waves

This section compares the experimental measurements and the numerical simulations for two cases with combined turbulent wind and irregular waves. The first case has a mean wind speed of 7.5 m/s with co-linear irregular waves with $H_s = 2.0$ m and

$T_p = 8.5\,\text{s}$. The second case has a mean speed of 11.4 m/s with co-linear irregular waves with $H_s = 3.0\,\text{m}$ and $T_p = 10.5\,\text{s}$. Two different numerical simulations are plotted against the experiments in this section. One of the simulations applies linear potential hydrodynamics ($openfast\ hyd : 1st$). The other one includes second order effects using the Newman approximation ($openfast\ hyd : 1st + 2nd$). For both, experiment and simulation results, it was used the same turbulent wind field, wave elevation time series and wind turbine controller.

Figure 9 and Fig. 10 shows the measured and simulated platform surge for the turbulent wind speed of 7.5 m/s and 11.4 m/s with their respective wave conditions. It can be seen that the numerical simulations are very close to the experimental response for both environmental conditions. Also, it can be noticed that the simulation including non linear hydrodynamics provides very similar results to the results of the linear model. This could be due to the relatively small height of the waves related with the significant wave height used. In the case of 11.4 m/s Fig. 10 both numerical solutions are also very similar. In this case

at rated wind speed, the platform motions are dominated by the wind load. The effect of second order hydrodynamics in the response of the platform under higher wave heights have been discussed by Azcona et al. (2019) and Roald et al. (2013).

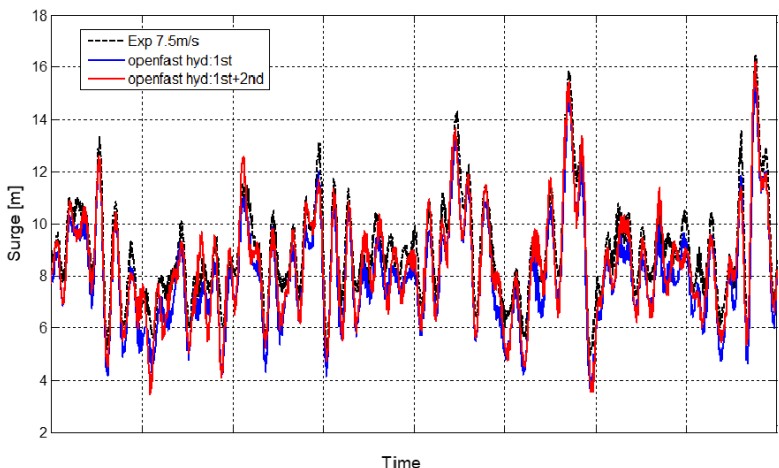

**Figure 9.** Time series for platform surge response under 7.5 m/s turbulent mean wind speed and irregular waves $H_s = 2.0\,\text{m}$; $T_p = 8.5\,\text{s}$

     The PSD of the surge responses for the experiments at 7.5 m/s and 11.4 m/s is presented in Fig. 11 (a) and (b) respectively. On both figures (a) and (b) the maximum energy is located at the platform surge natural frequency. This natural frequency is excited by low frequency loads such as the wind loading and the wave second order difference-frequency effects.

Fig. 11 (a) and (b) shows that the numerical platform surge motion with first order hydrodynamic is very similar to the experimental curve in the low frequency region. This indicates that with the wind and wave conditions tested the second order hydrodynamics is not contributing significantly to the platform response. This also mean that the wind turbine loading is dominating the response for both wind speed. This was also reported in Azcona et al. (2019) for the OC4 platform with the 5MW NREL wind turbine, where the wind loading dominates the platform response compared to hydrodynamics near rated wind

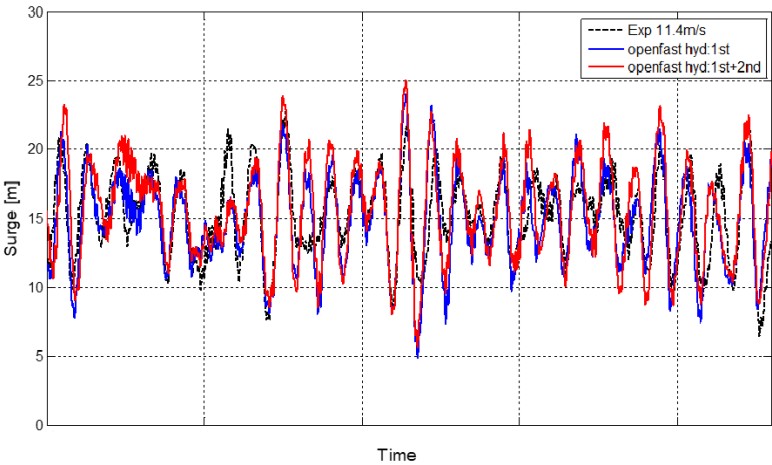

**Figure 10.** Time series for platform surge response under 11.4 m/s turbulent mean wind speed and irregular waves $H_s = 3.0$ m; $T_p = 10.5$ s

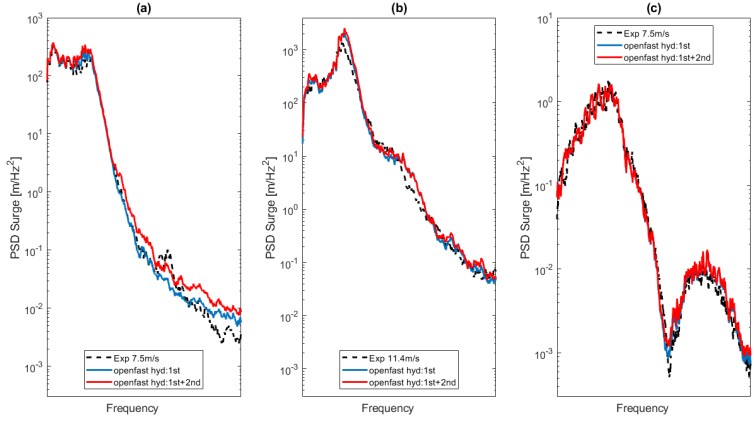

**Figure 11.** Surge PSD response of SATH10MW under (a) Turbulent wind with average of 7.5 m/s and irregular waves $H_s = 2.0$ m; $T_p = 8.5$ s, (b) Turbulent wind with average of 11.4 m/s and irregular waves $H_s = 3.0$ m; $T_p = 10.5$ s, (c) Wave region for turbulent wind with average of 7.5 m/s and irregular waves $H_s = 2.0$ m; $T_p = 8.5$ s

speed.

Fig. 11 (c) present a sudden decrease of the surge response inside the wave frequency region for the case of 7.5 m/s. This is a cancellation effect and is the result of the interaction of the length of the floater with the incident wave length that produce no force and moment on the SATH platform at a particular cancellation frequency. This effect is also detected in the surge

response at 11.4 m/s. This shows that 1st order hydrodynamics numeric modelling used can reproduce properly the dynamic

behaviour of the floater at the wave region.

The platform PSD pitch response for 7.5 m/s is presented in Fig. 12 (a). Both numerical models approach well the experimental response, but at the pitch natural frequency the numerical models underestimate the experimental peak. This might be caused by uncertainties in the coupling elements of the stiffness matrix that represents the retention system. The underprediction of low-frequency response has been also described in several publications such as Azcona et al. (2019) and Robertson et al. (2020). According to Robertson et al. (2020), the numerical model low-frequency response could be improved by including $2^{nd}$ order terms to the wave kinematics or tuning the drag coefficients.

The pitch response for the mean wind speed of 11.4 m/s is shown in Fig.12 (b). This PSD shows that the experimental pitch natural frequency is shifted to a lower frequency with respect to the value obtained from the free decay test and also with respect to the experiment for 7.5 m/s in Fig. 12 (a). The numerical results does not present this displacement to a lower frequency of the natural frequency, and the result is coherent with the natural frequency at the experiment for 7.5 m/s in Fig. 12 (a). The reason for this decrease in the experimental pitch natural frequency will be discussed in detail in the next section.

Figure 13 and Fig. 14 show that the platform yaw response predicted by the numerical simulation for the two turbulent wind speed match with very good agreement the experimental yaw response in time domain. This also indicates that the rotor moment around the vertical axis, $M_z$, introduced by the actuator in the experiment is correctly captured. The non-linear hydrodynamic is not producing a significant difference between in yaw numeric response, as it was seen for the surge and pitch responses. The platform yaw response is dominated by wind turbine loading introduced by means of "Mz" moment.

## 4.5 Hydrodynamic modeling of a floating platform in openFAST

As it was discussed in the previous section, in relation with in Fig. 12 (b) the platform pitch natural frequency in the experiment is shifted to a lower frequency, in comparison with the natural frequency observed in the free decays and in the PSD's for the cases with 7.5 m/s of turbulent mean wind speed.

This shift in the natural frequency could be caused by the change on the hydrostatic and hydrodynamic properties of the submerged substructure due to the pitch rotation of the platform at rated wind speed. In this case, the platform presents a mean pitch inclination of 4.3 deg and mean heave of -1.2 m.

To confirm this hypothesis, we built a new mesh of the submerged substructure for the geometry corresponding to the pitched platform at rated wind speed. Figure 15 (a) present the mesh for the original geometry of the submerged platform, used in WAMIT to calculate the added mass, potential damping, hydrostatic stiffness and wave excitation coefficients.

Figure 15 (b) present the mesh for the submerged substructure geometry corresponding to the mean pitch and heave at 11.4 m/s turbulent mean wind speed. It is equivalent to the platform position observed during the experiments as in Fig. 15 (c).

The new tilted geometry from Fig. 15 (b) produced a reduction in the hydrostatic stiffness for the pitch DoF of around 2%. The case with turbulent wind at rated wind speed of 11.4 m/s and irregular waves was simulated again with the new hydrostatic and hydrodynamic properties computed for the tilted geometry. The results are compared with the original simulation and

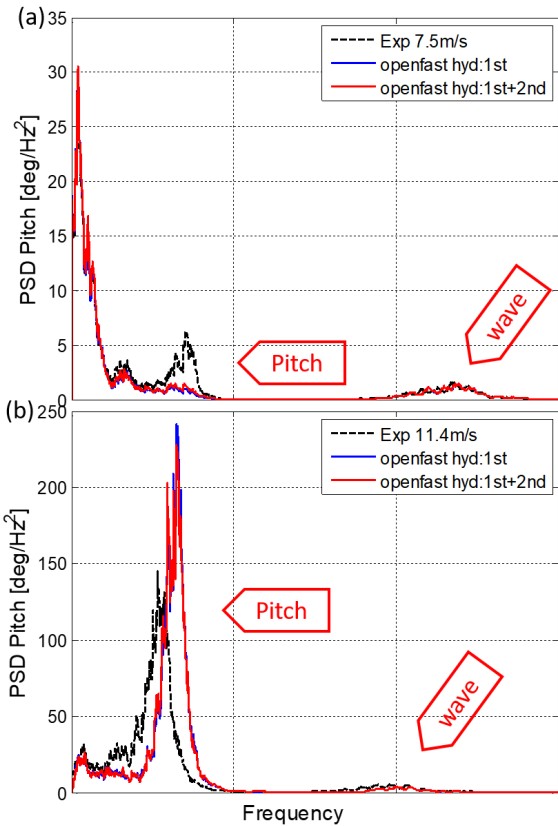

**Figure 12.** Pitch PSD response of SATH10MW under turbulent wind with average of (a) 7.5 m/s and irregular waves $H_s = 2.0$ m; $T_p = 8.5$ s and (b) 11.4 m/s and irregular waves $H_s = 3.0$ m; $T_p = 10.5$ s

with the experiments in the pitch PSD in Fig. 16. This plot shows that the natural frequency of numerical results with the tilted geometry also decreases with respect to the original simulation where the mesh is not pitched. The new frequency now matches the shifted pitch frequency from the experiments. This indicates that the advanced shape of this platform requires a careful consideration of the geometrical non-linearities to obtain accurate numerical results.

The pitch response at the wave frequency remained similar when the platform is tilted. Also, the surge response with the tilted geometry result was not different compared to the non-tilted geometry in the wave frequency region. The added mass in yaw increased a 14% with the tilted geometry, but it did not impact the platform yaw response.

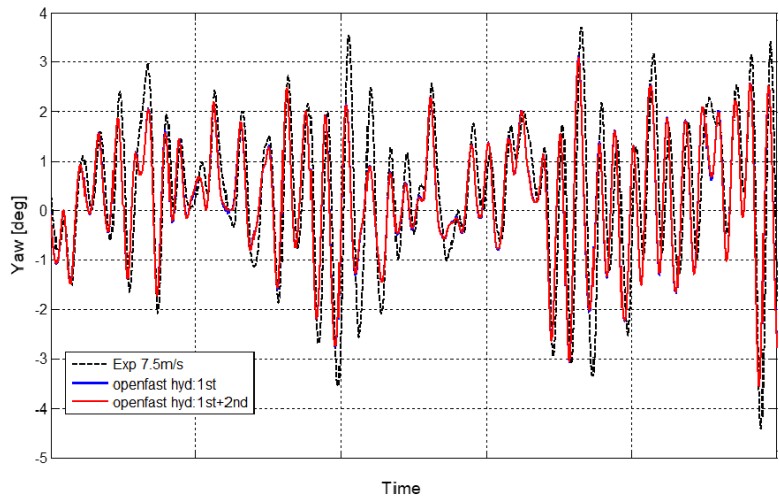

**Figure 13.** Yaw response of SATH10MW under turbulent wind with average of 7.5 m/s and irregular waves $H_s = 2.0\,\text{m}$; $T_p = 8.5\,\text{s}$

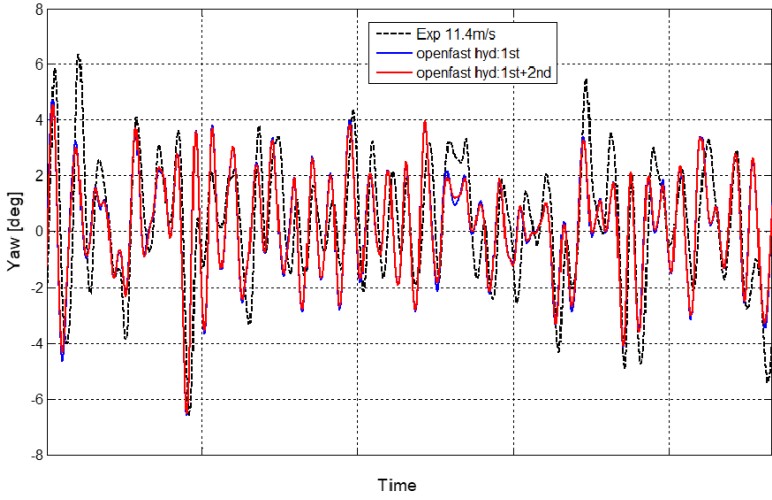

**Figure 14.** Yaw response of SATH10MW under turbulent wind with average of 11.4 m/s and irregular waves $H_s = 3.0\,\text{m}$; $T_p = 10.5\,\text{s}$

## 5   Conclusions

The hybrid SiL methodology was applied to a tank test campaign of the floating offshore substructure SATH supporting the INNWIND 10 MW wind turbine, which was performed at the Lir national Ocean Test Facility of UCC.

295

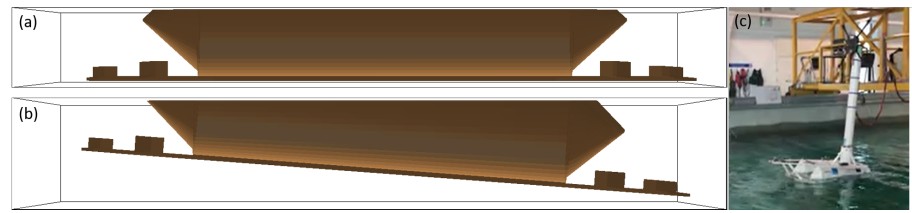

**Figure 15.** (a) Geometry of the wet surface of the platform at its equilibrium position. (b) Geometry of the wet surface of the platform considering the platform mean pitch and heave displacements under a turbulent wind of 11.4 m/s of mean wind speed and irregular wave. (c) Photograph of the platform during tank testing under wind and wave loading

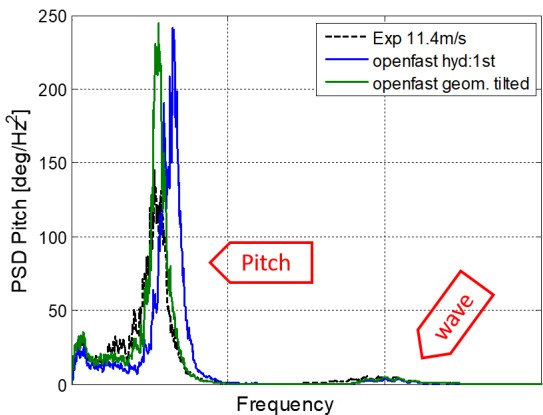

**Figure 16.** Pitch PSD response comparison between experiment curve, numeric model with platform without tilt and tilted under turbulent wind 11.4 m/s and irregular waves $H_s = 3.0$ m; $T_p = 10.5$ s

During the experimental campaign we used the most recent version of the SiL method developed at CENER that is able to introduce the rotor thrust and also the moments in the yaw and pitch axes. Therefore this enhanced SiL method including the out-of-plane rotor moment is able to reproduce the system's yaw motion. Good agreement between measurements and simulations were found for the platform motion in cases with steady wind only, turbulent wind only and simultaneous turbulent wind and irregular waves. In particular, the measured yaw response compares well with the simulations in OpenFAST for all the test cases considered, showing the successful performance of the new feature.

The wind turbine aerodynamic loading dominates the platform response under the wind and wave condition tested for the low frequencies. The second order hydrodynamics introduced in the simulations did not produce significant improvement of the numerical prediction for the conditions tested.

It was observed that for the case at rated wind speed there was an important shift in the pitch natural frequency of the experiment compared with experiment with lower wind loading. The simulation results do not capture this shift because the hydrostatic properties are not updated as the platform tilts. The simulation with the updated hydrostatic properties of the tilted geometry resulted also in a shift of the natural frequency that matched better the experimental results. This indicates that for a platform like SATH10MW with a complex platform geometry piercing the water plane area the hydrostatic stiffness is highly non-linear. The accurate numerical representation of the system dynamics might require to consider the variation of the hydrostatic stiffness matrix with the platform tilt.

*Author contributions.* FV reviewed the experimental data contributing to the results analysis, validation and calibrated the numerical model in OpenFAST. JA contributed in the conceptualization of the work, the experimental and validation results analysis, the preparation of the SiL hybrid numerical model and revision of the paper. OP supervised the SiL operation during tests at the tank and defined the actuator control system and improve the interface in LabVIEW of the SiL control program. IE developed the wind turbine controller for the SATH10MW INNWIND floating wind turbine. AR was responsible of the experiment tests and helped to define each tests of the experimental campaign. AM contribute in the definition of the experiment tests and model scales. CG was supervising the preparation activities before the test campaign. CD was responsible of the tank instrument and operation of the wave generator, also helped to define the test methodology. FV prepared the manuscript of this article with contribution from all co-authors

*Competing interests.* Authors declares that they have no conflict of interest

*Acknowledgements.* This test campaign and the results presented were developed within the project ARCWIND – Adaptation and implementation of floating wind energy conversion technology for the Atlantic region, which is co-financed by the Interreg Atlantic Area Programme through the European Regional Development Fund under contract EAPA 344/2016. Authors want also to give thanks for their assistance during the experiments set up to Christian van den Bosch and Otter Aldert from University College of Cork. Also, authors want to thanks Juan Martínez Belio and Jon Olagüe from the Structural Area of Dept. Wind turbine analysis and design (CENER) for his support on the mesh generation from the CAD floating platform.

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
