# Peer review of "Model tests of a 10 MW semi-submersible floating wind turbine under waves and wind using hybrid method to integrate the rotor thrust and moments"

_Wind Energy Science, 2021_

## Referee Comment (RC1)

Review of "Model test of a 10MW semi-submersible floating wind turbine under waves and wind using hybrid method to integrate the rotor thrust and moments" by F. Vittori et al.

The authors present results from an experimental campaign on a scaled 10MW wind turbine model. The turbine is mounted on a floater designed by Saitec. Rotor forces are modelled with a SiL approach using 4 propellers. Experimental data is compared to a numerical OpenFAST model, and excellent agreement is found. Results appear credible and the methods seem solid. This said, I have some comments:

From a general standpoint, the methods section can be greatly expanded. More details regarding the mooring system are required. No specific details are presented regarding the floater. Also the design choices should be presented in more detail. For instance, what are the design objectives of this particular floater-tower design?

My main concern regards the scope of this paper, that should be addressed clearly. Since the results and the testcase used in the study appear to not be publicly available as they are industrial IP, the scope of this work is not obvious. The original contributions should be better highlighted and conclusions should highlight how industry and academia can exploit the results of this work. I believe the paper currently lacks these considerations.

Below some more detailed considerations:

P3 L60: Please clarify: "This paper shows how the SiL method, including rotor moments, is able to reproduce the dynamic response of the SATH 10MW INNWIND (SATH10MW) floating offshore wind turbine." How is it shown that the SiL method is able to reproduce the dynamic response?

P4 L70 "The scaled thrust is controlled by the motor rotational speed set by an electronic controller, which again depends on the real time simulation of the full scale rotor in a turbulent wind field, considering turbine control action with the platform motions measured in real time in the wave tank test." This is not clear, please rephrase.

P5 L95: Glauert correction is a correction for high induction wake states, the phrase " The aerodynamic loads are based in Blade Element Momentum (BEM) model using the Glauert correction." is confusing. Authors state that the blades and tower are considered rigid, what is the rationale here? This assumption is not always the case for model tests (in OC5 for instance a flexible tower was considered in the numerical models). Please expand.

P5 L102: Why is the SPM mooring system not implemented? And what are the expected impact on system dynamics of the included mooring system? Please expand.

P5 L97: Consider expanding on the way the WT controller was designed, this is an important part of any WT, especially a floating one. Why was a custom controller developed? Is it tailored to the specific needs of this floater design?

Section 3: No mention on how the mooring system is modelled is presented.

P8 L155: It is not clear how the numerical model was calibrated to account for the cable bundle in the experiments, please clarify.

Figures 7-16: When PSDs are presented no labels are included on x-axis, to protect industrial IP. I suggest to normalize the x-values by some physical parameter. A good choice could be the surge natural frequency.

P12 L223: Do the authors have an explanation on why pitch response (fig 12a) at the systems natural frequency seems to be almost completely missed by the numerical models?

Conclusions: Besides the good agreement between numerical and experimental data, the innovative contribution of this study is not clear. Please include some outlook on the relevance of these results.

---

## Referee Comment (RC2)

I have reviewed the paper entitled "Model tests of a 10 MW semi-submersible floating wind turbine under waves and wind using hybrid method to integrate the rotor thrust and moments" by Felipe Vittori et al. and submitted to Wind Engineering Science.

The authors describe an experimental campaign testing a floating offshore wind turbine by means of an hybrid method. They manage to generate wave and wind loadings. The novelty of the setup is that they includes not only the main aerodynamic thrust but also moments in the turbine plane, around the vertical and horizontal axis. As a main result, the authors shows that in order to predict the pitch natural period, the hydrodynamic database should be computed based on the geometry of the floater in its mean position (tilt) due to the average wind loading. This result is surely general to all the floater that have significant change of hydrostatic restoring moment with trim.

The floater they used in the campaign is developed by Saitec. This restricts the data that can be shared like the natural periods, the size of the platform, the description of the mooring lines.

The development of the SiL system is described in previous papers. The main components of the setup are briefly described in the paper (actuators, software for aerodynamic computations, software for actuator command). The performance of the system for the frequency range at aim in this paper (from 0 to 0.16Hz FS, 0 to 1.2Hz MS) is missing (phase, gain, response time, frequency bandwidth and delays); such data should be used in the discussion of the comparisons between experiments and simulations. Is there a load cell at the top of the mast that could provide the measured thrust and moments? That would be nice to compare the aerodynamic loads in the experiments (imposed by the SiL system) and the loads in the simulations, in the time domain and in the frequency.

One main comment is that the authors present many observations of the results shwon in the Figures but they often stop there. They should always try to give possible explanations or ideas that would help understand the origin of such observations.

The authors used confidential restrictions in the decay tests (cf. line 161), while in the other tests, the presentation of results is not so strict and the given results enable the reader to find some data that might be confidential.
For instance, the reader can deduce the natural periods from the time domain Figures 7a, 7c, 8a, 8c (time axis is given explmicitly with figures and unit) and the frequency domain results presented in Figures 7b, 7d and 8b, 8d may be used to confirm those, even if the frequency axis is given without figures. Here are some guess, at FS
Surge natural period is estimated at 86 s (12 mHz).
Sway natural period is estimated at 106 s (9.5 mHz).
Pitch natural period is estimated at 30 s (33 mHz).
Yaw natural period is estimated at 63 s (16 mHz).

Remarks:

paragraph 2.2
The experimental setup matchs the target RNA mass. What about the CoG position (important for trim angle) and moments of inertia (less important, for dynamics).

Regarding the calibration of the numerical model, it seems that it consisted in tuning the stiffness matrix and the damping coefficients. The corresponding paragraphs are however in different

sections. Concerning the stiffness that was modified during decay test, how was it obtained in the first place? theoretically?

Yaw decay test shows the worst "good agreement" when compared to the other presented DoF decays. It would worth mentionning the main possible causes, the efforts taken to reduce that and the consequences on the subsequent comparisons between experimental and numerical results such as section 4.3 and 4.4.

Have some decay tests with wind been performed? For instance with constant wind speed, the effect of aerodynamic damping could have observed and compared in simulation and experiments.

The note on the surge and sway definitions (line 160) should be moved earlier when the Figure 4 is described.

Figure 7,
The experimental surge motion is very well reproduced by the numerical model. The relative error on the moment of order zero of the PSD must be very small.
The same agreement is found for pitch motion at low frequency but around pitch natural frequency, the numerical model estimation is lower than the experimental one. What is the performance of the SiL system around that frequency? In other words, is the experimental peak at 33 mHz an expected feature or is it due to the SiL system interacting with the resonant DoF?

Figure 8,
In this Figure, the numerical responses differ from the experimental ones. Possible reasons are missing in the paper.
In sway, the Fy force is not implemented by the SiL system. Is it modeled numerically or is the aerodynamic loads limited to the thrust only, like in the experiments?
The yaw response being larger in the experiments, the projection of the thrust will have a bigger impact on sway, however the difference between the significant yaw angle in the experiments and in the simulation is not large enough to generate such differences on the sway motion.
What about the cable bundle? Is it inline with the surge axis or is it pulling sideway too?
Is the Fy force generated by the multi-propeller system negligible?

Figures 9 and 10,
The effort made to include 2nd hydrodynamic wave forces in the numerical model deserve more than a statement as simple as "certain improvment". The differences between the two numerical models should be quantified. The world "probably" can be removed (line 207). The second-order forces being nonlinear by definition, more severe sea-states would be more interesting if one want to see the effects of such loads with respect to the wind loads, at low frequency.

Results, Figure 11,
Cancellation happens at a wave period of 6.3 s (160 mHz) at FS. It would be meaningfull to compare numerically the corresponding wavelength (61 m) to the length of the floater. The study of the magnitude of the excitation forces given by WAMIT in surge would confirm (if needed) the position of the cancellation frequency. The fact that the numerical model agrees well with the experiment means that the excitation force and the added-mass are correctly computed. I refer here at the added mass since at such high frequency, the RAO is indeed the ratio of excitation force divided by mass and added-mass terms.

Figure 13 and 14,
What is the reason for keeping 2 numerical models here? If any, it is not given in the next where no mention to the hydrodynamic models appears.

The agreement is no doubt very good, in particular when compared to SiL systems that don't offer this capability of generating moments. Looking in more details however, we see both an amplitude mismatch of the yaw amplitude and a delay, at wind speed 7.5 m/s.

If the SiL system is to be used for a FOWT that presents a mooring yaw stiffness, then the effect of the vertical axis moment will have to be quantified.

Section 4.5,

The results presented in this section are of great interest for the reader willing to study floating wind turbines. The hydrodynamic software used in the paper have been developped, validated and used for offshore oil and gas systems where the rest position of the floater is the mean position in waves, and where the effect of wind is mainkly an additional drift force.

For floating wind turbine, the wind has one major contribution, as said by the authors: the mean geometry of the floater is changed due the trim generated by the mean aero thrust and this may have consequences on the hydrodynamic forces such as the hydrostatic stiffness.

The response in pitch is overstimated by the numerical simulation wrt the experiments. What modification of the pitch damping coefficients would be necessary to catch the correct response magnitude at the pitch natural frequency? Would such a modification affects the good agreement observed at periods longer than the pitch natural period?

The SiL performance may also contribute in terms of gain and delay, although the considered periods may be large compared to the response time of the SiL system.

---

## Author Response (AR1)

**Response to the editor**

2nd July 2022

Dear Editor,

This document indicates the comments from RC1 and RC2, our response and where these comments were incorporated in the new version of the document.

**Response to RC1 comments**

*Comments to the authors:*
The authors present results from an experimental campaign on a scaled 10MW wind turbine model. The turbine is mounted on a floater designed by Saitec. Rotor forces are modelled with a SiL approach using 4 propellers. Experimental data is compared to a numerical OpenFAST model, and excellent agreement is found. Results appear credible and the methods seem solid. This said, I have some concerns regarding the scientific impact of this paper, which is not stated clearly. The original contributions of this work must be stated more clearly so that the scientific community can benefit from them. Some possible suggestions:

- better highlight some of the methods that were found to be effective in improving agreement between experimental and numerical approaches.

- provide additional analysis to better highlight open issues and provide outlook on how what would be needed to solve them.

More detailed comments in the attached pdf.

*Response to comments to the Authors:*
The authors appreciate the comments from RC1. The suggestions are addressed through the detailed comments below.

*Comments to the authors:*
The authors present results from an experimental campaign on a scaled 10MW wind turbine model. The turbine is mounted on a floater designed by Saitec. Rotor forces are modelled with a SiL approach using 4 propellers. Experimental data is compared to a numerical OpenFAST model, and excellent agreement is found. Results appear credible and the methods seem solid. This said, I have

some comments:

*Comment RC1 - No. 1:*
From a general standpoint, the methods section can be greatly expanded. More details regarding the mooring system are required. No specific details are presented regarding the floater. Also the design choices should be presented in more detail. For instance, what are the design objectives of this particular floater-tower design?

*Response to comment RC1 - No. 1:*
Additional description about SATH design were included in line 25 and 120 followed by the address to the web page of Saitec Technologies to know learn more about the floater design.

*Comment RC1 - No. 2:*
My main concern regards the scope of this paper, that should be addressed clearly. Since the results and the testcase used in the study appear to not be publicly available as they are industrial IP, the scope of this work is not obvious. The original contributions should be better highlighted and conclusions should highlight how industry and academia can exploit the results of this work. I believe the paper currently lacks these considerations.

*Response to comment RC1 - No. 2:*
In this work we want to highlight two main outcomes. The first one is that we have tested a new feature in the SiL hybrid method: including the wind turbine moments (not only the thrust). Due to this improvement, the platform response in yaw, induced by the wind turbine out-of-plane moment around the vertical moment "Mz" was obtained. In the subsequent validation of the numeric model in OpenFAST of the SATH 10MW model in full scale it was observed a good agreement between numerical results and experimental measurements of the platform yaw response under the same wind and wave conditions. This shows the good performance of the new feature of the SiL method. The second relevant outcome is related with a variation of the platform pitch natural frequency obtained in the experiments under high rotor thrust, compared to the natural frequency obtained in free decays. This variation is caused by a change of the hydrostatic stiffness when the platform tilts. During validation of numerical model it was observed that OpenFAST was not able to capture this variation in the natural frequency. The reason is that OpenFAST assumes a linear behavior of the hydrostatic stiffness matrix that is obtained at the un-displaced platform position. Nevertheless, the complex geometry of SATH platform produces a highly non-linear behavior of this stiffness. For this reason, once the stiffness matrix was recomputed for the tilted platform, the pitch natural frequency of the computation agrees with the experiments. This limitation of the common linear potential hydrodynamic numerical model is an open issue. A better computation of the instantaneous buoyancy loads may improve the simulation of dynamic response of a FOWT with this type of floater geometry.

Action on Manuscript: Introduction was modified at line 64 to 67. Conclusion was modified.

*Comment RC1 - No. 3:*

P3 L60: Please clarify: "This paper shows how the SiL method, including rotor moments, is able to reproduce the dynamic response of the SATH 10MW IN-NWIND (SATH10MW) floating offshore wind turbine." How is it shown that the SiL method is able to reproduce the dynamic response?

*Response to comment RC1 - No. 3:*

Action on Manuscript: Sentence was removed and paragraph modified, now in line 64.

*Comment RC1 - No. 4:*

P4 L70 "The scaled thrust is controlled by the motor rotational speed set by an electronic controller, which again depends on the real time simulation of the full scale rotor in a turbulent wind field, considering turbine control action with the platform motions measured in real time in the wave tank test." This is not clear, please rephrase.

*Response to comment RC1 - No. 4:*

The scaled thrust is controlled by an electronic controller (EC) that regulates the rotational speed of the propellers motor of the actuator. This EC receives the thrust demand from a real time full scale simulation of the wind turbine. The simulation takes into account the wind field, the wind turbine control and the real time platform motions measured in the wave tank. Therefore, the method captures the coupling between the rotor loads and the platform motions, which is a relevant effect to accurately represent the dynamics of a floating wind turbine.

Action on Manuscript: This paragraph was included and it is now at 76

*Comment RC1 - No. 5:*

P5 L95: Glauert correction is a correction for high induction wake states, the phrase " The aerodynamic loads are based in Blade Element Momentum (BEM) model using the Glauert correction." is confusing. Authors state that the blades and tower are considered rigid, what is the rationale here? This assumption is not always the case for model tests (in OC5 for instance a flexible tower was considered in the numerical models). Please expand.

*Response to comment RC1 - No. 5:*

The tower for the experimental scaled model (Fig. 3) was designed rigid, with a larger diameter to avoid any elastic response. The numerical model used in the hybrid testing is fed by the motions measured at the intersection between the tower centerline and the water plane. For consistency with the scaled model, this numerical model assumed a rigid tower. The blades of the numerical model are also assumed rigid to improve the CPU speed ensuring real time, and because the loss of accuracy is low compared to other sources of uncertainty of the

experimental setup.

Action on Manuscript: L95 removed.

*Comment RC1 - No. 6:*

P5 L102: Why is the SPM mooring system not implemented? And what are the expected impact on system dynamics of the included mooring system? Please expand.

*Response to comment RC1 - No. 6:*

In this test campaign the SPM system was not implemented to ease the uncertainties on the initial validation of the numerical tools and floating design. The retention system constrains all the platform degree of freedom. In contrast, once the SPM is enabled, the platform will be able to freely rotate in yaw offering a different dynamics of the platform.

*Comment RC1 - No. 7:*

P5 L97: Consider expanding on the way the WT controller was designed, this is an important part of any WT, especially a floating one. Why was a custom controller developed? Is it tailored to the specific needs of this floater design?

*Response to comment RC1 - No. 7:*

Action on Manuscript: Description expanded from L 111 to L 117.

*Comment RC1 - No. 8:*

Section 3: No mention on how the mooring system is modelled is presented.

*Response to comment RC1 - No. 8:*

The mooring system was modelled using a stiffness matrix (6x6) in Hydrodyn with coupled terms at the respective DoF. First, the matrix coefficients were defined mathematically and afterwards the coefficients were tuned according to the platform response in the free decays.

Action on Manuscript: Description added L150

*Comment RC1 - No. 9:*

P8 L155: It is not clear how the numerical model was calibrated to account for the cable bundle in the experiments, please clarify.

*Response to comment RC1 - No. 9:*

From experimental pitch free decays with and without cable bundle. We estimated a 3% of difference in platform pitch stiffness. The numeric model included this effect by including an additional pitch stiffness coefficient. Additionally, a preload moment was required to match the pitch mean position.

Action on Manuscript: Description added L183

*Comment RC1 - No. 10:*

Figures 7-16: When PSDs are presented no labels are included on x-axis, to

protect industrial IP. I suggest to normalize the x-values by some physical parameter. A good choice could be the surge natural frequency.

*Response to comment RC1 - No. 10:*
Action on Manuscript: Figures modified accordingly.

*Comment RC1 - No. 11:*
P12 L223: Do the authors have an explanation on why pitch response (fig 12a) at the systems natural frequency seems to be almost completely missed by the numerical models?

*Response to comment RC1 - No. 11:*
We have discarded that the difference in the PSD peaks are caused by low frequency 2nd order effects because this difference also appears at the turbulent wind only cases. Thus, we believe that the difference in PSD pitch peaks are related with uncertainties in the characterization of the couplings in DoF of the retention systems and the changes of the hydrostatic coefficient in pitch that are occurring in the experiments but not modeled in OpenFAST.

*Comment RC1 - No. 12:*
Conclusions: Besides the good agreement between numerical and experimental data, the innovative contribution of this study is not clear. Please include some outlook on the relevance of these results.

*Response to comment RC1 - No. 12:*
Action on Manuscript: Conclusions were modified accordingly and it was taken into account the response for RC1 No.1 comment.

**Response to RC2 comments**

*Comments to the authors:*
The authors describe an experimental campaign testing a floating offshore wind turbine by means of an hybrid method. They manage to generate wave and wind loadings. The novelty of the setup is that they includes not only the main aerodynamic thrust but also moments in the turbine plane, around the vertical and horizontal axis. As a main result, the authors shows that in order to predict the pitch natural period, the hydrodynamic database should be computed based on the geometry of the floater in its mean position (tilt) due to the average wind loading. This result is surely general to all the floater that have significant change of hydrostatic restoring moment with trim.

The floater they used in the campaign is developed by Saitec. This restricts the data that can be shared like the natural periods, the size of the platform, the description of the mooring lines.

*Comment RC2 - No. 1:*
The development of the SiL system is described in previous papers. The main

components of the setup are briefly described in the paper (actuators, software for aerodynamic computations, software for actuator command). The performance of the system for the frequency range at aim in this paper (from 0 to 0.16Hz FS, 0 to 1.2Hz MS) is missing (phase, gain, response time, frequency bandwidth and delays); such data should be used in the discussion of the comparisons between experiments and simulations. Is there a load cell at the top of the mast that could provide the measured thrust and moments? That would be nice to compare the aerodynamic loads in the experiments (imposed by the SiL system) and the loads in the simulations, in the time domain and in the frequency.

*Response to comment RC2 - No. 1:*
The inclusion of a load cell, to measure force and moment, in the force actuator is an improvement that we are working on. For this tank test we could not fit the load cell in the force actuator. However, we run calibration tests on the actuator, before send the device to the tank, to check the response of the motors and use signal filtering to avoid interruptions.

*Comment RC2 - No. 2:*
One main comment is that the authors present many observations of the results shown in the Figures but they often stop there. They should always try to give possible explanations or ideas that would help understand the origin of such observations.

*Response to comment RC2 - No. 2:*
Comment acknowledged and implemented across the new version of the document.

*Comment RC2 - No. 3:*
The authors used confidential restrictions in the decay tests (cf. line 161), while in the other tests, the presentation of results is not so strict and the given results enable the reader to find some data that might be confidential. For instance, the reader can deduce the natural periods from the time domain Figures 7a, 7c, 8a, 8c (time axis is given explicitly with figures and unit) and the frequency domain results presented in Figures 7b, 7d and 8b, 8d may be used to confirm those, even if the frequency axis is given without figures. Here are some guess, at FS: Surge natural period is estimated at 86 s (12 mHz). Sway natural period is estimated at 106 s (9.5 mHz). Pitch natural period is estimated at 30 s (33 mHz). Yaw natural period is estimated at 63 s (16 mHz).

*Response to comment RC2 - No. 3:*
Comment noted. The respective figures were adjusted.

*Comment RC2 - No. 4:*
paragraph 2.2 The experimental setup matches the target RNA mass. What about the CoG position (important for trim angle) and moments of inertia (less important, for dynamics).

*Response to comment RC2 - No. 4:*
The equilibrium pitch for this floating platform was below -1deg, from the numeric estimation. The RNA mass of the scaled model was adjusted taking care of not exceeding this value. The COG location and the moments of inertia were calculated based on numerical mass distribution calculations. We added significant amounts of lead to the heave plate, the transition piece and the nacelle to achieve the correct distribution. The difference in the MOI is below 1% and the CoG position is below 5mm in any of the 3 directions.

Action on Manuscript: This review was included in line 135.

*Comment RC2 - No. 5:*
Regarding the calibration of the numerical model, it seems that it consisted in tuning the stiffness matrix and the damping coefficients. The corresponding paragraphs are however in different sections.

*Response to comment RC2 - No. 5:*
The calibration process is not presented just the final results that are shown together with the free decay results. A reference is added to link with calibration results section.

Action on Manuscript:
Linear and quadratic damping coefficients of the numeric model were calibrated through experimental decay test results as it is shown in section 4.1.
Text added in line 148

*Comment RC2 - No. 6:*
Concerning the stiffness that was modified during decay test, how was it obtained in the first place? theoretically?

*Response to comment RC2 - No. 6:*
The linear stiffness matrix that represent the mooring system was initially defined analytically according the mooring lines tension, fairlead positions and the scaled model geometry. The coefficient of the matrix were later adjusted to match the experimental results.

Action on Manuscript: text added in line 150

*Comment RC2 - No. 7:*
Yaw decay test shows the worst "good agreement" when compared to the other presented DoF decays. It would worth mentioning the main possible causes, the efforts taken to reduce that and the consequences on the subsequent comparisons between experimental and numerical results such as section 4.3 and 4.4.

*Response to comment RC2 - No. 7:*

The numeric yaw response presents a constant oscillation period that was adjusted to match the first oscillations periods from the experimental decay. We consider that the shift in experimental yaw period is due to uncertainties in the estimation of yaw stiffness in the seakeeping system.

Action on Manuscript: Explanation modified accordingly in line 188

*Comment RC2 - No. 8:*
Have some decay tests with wind been performed? For instance with constant wind speed, the effect of aerodynamic damping could have observed and compared in simulation and experiments.

*Response to comment RC2 - No. 8:*

We have performed decay tests with wind SiL system to ensure the correct performance of the hybrid system. Nevertheless, for the paper we have focused on the turbulent wind only case that is another form of observe the aerodynamic damping from the wind turbine and the controller actions but under a more complex condition.

*Comment RC2 - No. 9:*
The note on the surge and sway definitions (line 160) should be moved earlier when the Figure 4 is described.

*Response to comment RC2 - No. 9:*
The note, is now at line 180 as suggested.

*Comment RC2 - No. 10:*
Figure 7, The experimental surge motion is very well reproduced by the numerical model. The relative error on the moment of order zero of the PSD must be very small. The same agreement is found for pitch motion at low frequency but around pitch natural frequency, the numerical model estimation is lower than the experimental one. What is the performance of the SiL system around that frequency? In other words, is the experimental peak at 33 mHz an expected feature or is it due to the SiL system interacting with the resonant DoF?

*Response to comment RC2 - No. 10:*

We do not consider that the pitch peak from experiments is related with a resonance because the peak amplitude would be much larger than the one obtained. Instead we see that the difference in PSD curves around pitch natural frequencies are related with changes in the water plane area of the scaled model that varies the pitch natural frequency. The Hydrodyn numeric model uses constant hydrostatic coefficients that are limiting the numeric response of the platform.

*Comment RC2 - No. 11:*
Figure 8, In this Figure, the numerical responses differ from the experimental ones. Possible reasons are missing in the paper. In sway, the Fy force is not implemented by the SiL system. Is it modeled numerically or is the aerodynamic loads limited to the thrust only, like in the experiments? The yaw response being larger in the experiments, the projection of the thrust will have a bigger impact on sway, however the difference between the significant yaw angle in the experiments and in the simulation is not large enough to generate such differences on the sway motion. What about the cable bundle? Is it inline with the surge axis or is it pulling sideway too? Is the Fy force generated by the multi-propeller system negligible?

*Response to comment RC2 - No. 11:*

The Fy from the rotor is included in the numeric model. The rotor is fully simulated in OpenFast producing the respective aerodynamic 3 forces and 3 moment. The multi-propeller actuator only can generate forces in the direction showed in Figure 3. It cannot produce forces in the perpendicular directions "Fy" and "Fz". The sway motion is relatively small compared with the surge motion and could be produced by the difference between experimental and numerical model. The yaw motion could be due to a coupling of the sway-surge degrees of freedom with yaw through the seakeeping system. The bundle cable was not constrained to move in any direction. We have accounted it effect during the decay test but for this cases is difficult to determine its influence. Certainly, it should not be discarded. We are working in the reduction of number of cables to control the actuator and reduce any other bias in the experiment.

*Comment RC2 - No. 12:*
Figures 9 and 10, The effort made to include 2nd hydrodynamic wave forces in the numerical model deserve more than a statement as simple as "certain improvement". The differences between the two numerical models should be quantified. The world "probably" can be removed (line 207). The second-order forces being nonlinear by definition, more severe sea-states would be more interesting if one want to see the effects of such loads with respect to the wind loads, at low frequency.

*Response to comment RC2 - No. 12:*

After considering the differences between the simulations with second order effects and just linear, we believe that the differences are small due to the low height of the waves. We have updated the text accordingly.
    Action on Manuscript: text updated from line 232.

*Comment RC2 - No. 13:*
Results, Figure 11, Cancellation happens at a wave period of 6.3 s (160 mHz) at FS. It would be meaningfull to compare numerically the corresponding wavelength (61 m) to the length of the floater. The study of the magnitude of the excitation forces given by WAMIT in surge would confirm (if needed) the position of the cancellation frequency. The fact that the numerical model agrees well with the experiment means that the excitation force and the added-mass are correctly computed. I refer here at the added mass since at such high frequency, the RAO is indeed the ratio of excitation force divided by mass and added-mass terms.

*Response to comment RC2 - No. 13:*

This is a very interesting comment and we will study it to get a better understanding of the experimental results for future analysis and publications. Nevertheless, the deadline for the revision of the paper are tight and we do not have time enough to include this analysis with the required background analysis in time.

*Comment RC2 - No. 14:*
Figure 13 and 14, What is the reason for keeping 2 numerical models here? If any, it is not given in the next where no mention to the hydrodynamic models appears.

The agreement is no doubt very good, in particular when compared to SiL systems that don't offer this capability of generating moments. Looking in more details however, we see both an amplitude mismatch of the yaw amplitude and a delay, at wind speed 7.5 m/s. If the SiL system is to be used for a FOWT that presents a mooring yaw stiffness, then the effect of the vertical axis moment will have to be quantified.

*Response to comment RC2 - No. 14:*

We want to show the effect of the non-linear hydrodynamics in all the DoF of the platform. The platform yaw response is dominated by wind turbine loading introduced by means of "Mz" moment.

Justification added in line 265.

*Comment RC2 - No. 15:*
Section 4.5, The results presented in this section are of great interest for the reader willing to study floating wind turbines. The hydrodynamic software used in the paper have been developed, validated and used for offshore oil and gas systems where the rest position of the floater is the mean position in waves, and where the effect of wind is mainly an additional drift force. For floating wind turbine, the wind has one major contribution, as said by the authors: the mean geometry of the floater is changed due the trim generated by the mean aero thrust and this may have consequences on the hydrodynamic forces such as the hydrostatic stiffness. The response in pitch is overestimated by the numerical simulation wrt the experiments. What modification of the pitch damping coefficients would be necessary to catch the correct response magnitude at the pitch natural frequency? Would such a modification affects the good agreement

observed at periods longer than the pitch natural period? The SiL performance may also contribute in terms of gain and delay, although the considered periods may be large compared to the response time of the SiL system.

*Response to comment RC2 - No. 15:*

To improve the FOWT response is important to take into account non-linear buoyancy effect in the numeric model. This allows to use variable coefficient in the hydrostatic stiffness matrix that represent better the floater during it dynamic response. To propose new damping implementations in numeric modelling it is required to study more in detail the damping sources of this particular floater design, considering the effect of the geometry trim. Regarding the SiL method, the development of the technology is continued not only to improve the wind turbine loading scaled representation but also in state parameters to measure the actuator performance.

---

## Referee Report (RR1)

The manuscript has improved and the authors have adequately addressed most of the comments. Some minor remarks still remain unaddressed and are reported in the following:

- A general revision of the English is recommended. For example, line 301 "produce" not "produced".

- The following comment was not addressed in the manuscript:
  Comment RC1- No. 11: P12 L223: Do the authors have an explanation on why pitch response (fig 12a) at the systems natural frequency seems to be almost completely missed by the numerical models?
  Response to comment RC1- No. 11: We have discarded that the difference in the PSD peaks are caused by low frequency 2nd order effects because this difference also appears at the turbulent wind only cases. Thus, we believe that the difference in PSD pitch peaks are related with uncertainties in the characterization of the couplings in DoF of the retention systems and the changes of the hydrostatic coefficient in pitch that are occurring in the experiments but not modeled in OpenFAST.
  Please include this hypothesis in the manuscript, it could help other reaserchers that are observing the same. Also, underprediction of low-frequency response is also observed in various literature works, that may help explain what you observed: [1,2]
  [1]     Robertson A N, Gueydon S, Bachynski E, Wang L, Jonkman J, Alarcón D, Amet E, Beardsell A, Bonnet P, Boudet B, Brun C, Chen Z, Féron M, Forbush D, Galinos C, Galvan J, Gilbert P, Gómez J, Harnois V, Haudin F, Hu Z, Dreff J L, Leimeister M, Lemmer F, Li H, Mckinnon G, Mendikoa I, Moghtadaei A, Netzband S, Oh S, Pegalajar-Jurado A, Nguyen M Q, Ruehl K, Schünemann P, Shi W, Shin H, Si Y, Surmont F, Trubat P, Qwist J and Wohlfahrt-Laymann S 2020 OC6 Phase I: Investigating the underprediction of low-frequency hydrodynamic loads and responses of a floating wind turbine *J. Phys.: Conf. Ser.* **1618** 032033
  [2]     Wang L, Robertson A, Jonkman J and Yu Y-H 2022 OC6 phase I: Improvements to the OpenFAST predictions of nonlinear, low-frequency responses of a floating offshore wind turbine platform *Renewable Energy* **187** 282–301

- **Figures 7-16:** No values are present in the x-axis. Please add labels to the x-axis of the PSDs if possible. I suggest normalizing the x-values by some physical parameter. A good choice could be the surge natural frequency.

---

## Author Response (AR2)

**Response to reviews**

**September 8, 2022**

Thank you for the revision of our manuscript. In this document we indicate the response to the referee's comments and how we have included them into the paper.

**Referee #1:**

The manuscript has improved and the authors have adequately addressed most of the comments. Some minor remarks still remain unaddressed and are reported in the following:

A general revision of the English is recommended. For example, line 301 "produce" not "produced".

We have reviewed the English and we have corrected: "hypothesis" instead of "hypotheses" (Line 277), "produce" instead of "produced" (Line 301), "are showed" instead of "are shown" (abstract)

The following comment was not addressed in the manuscript: Comment RC1- No. 11: P12 L223: Do the authors have an explanation on why pitch response (fig 12a) at the systems natural frequency seems to be almost completely missed by the numerical models? Response to comment RC1- No. 11: We have discarded that the difference in the PSD peaks are caused by low frequency 2nd order effects because this difference also appears at the turbulent wind only cases. Thus, we believe that the difference in PSD pitch peaks are related with uncertainties in the characterization of the couplings in DoF of the retention systems and the changes of the hydrostatic coefficient in pitch that are occurring in the experiments but not modeled in OpenFAST. Please include this hypothesis in the manuscript, it could help other reaserchers that are observing the same. Also, underprediction of low-frequency response is also observed in various literature works, that may help explain what you observed: [1,2] [1] Robertson A N, Gueydon S, Bachynski E, Wang L, Jonkman J, Alarcón D, Amet E, Beardsell A, Bonnet P, Boudet B, Brun C, Chen Z, Féron M, Forbush D, Galinos C, Galvan J, Gilbert P, Gómez J, Harnois V, Haudin F, Hu Z, Dreff J L, Leimeister M, Lemmer F, Li H, Mckinnon G, Mendikoa I, Moghtadaei A, Netzband S, Oh S, Pegalajar-Jurado A, Nguyen M Q, Ruehl K, Schünemann P, Shi W, Shin H, Si Y, Surmont F, Trubat P, Qwist J and Wohlfahrt-Laymann S 2020 OC6 Phase I: Investigating the underprediction of low-frequency hydrodynamic loads and responses of a floating wind turbine J. Phys.: Conf. Ser. 1618 032033 [2] Wang L, Robertson A, Jonkman J and Yu Y-H 2022 OC6 phase I: Improvements to the OpenFAST predictions of nonlinear, low-frequency responses of a floating offshore wind turbine platform Renewable Energy 187 282–301

We have cited the mentioned reference and add a brief discussion on why, according to the current knowledge, this underestimation could be caused: *"The underprediction of low-frequency response has been also described in several publications such as \cite{azcona19} and \cite{robertson20}. According to \cite{robertson20}, the numerical model low-frequency response could be improved by including $2^{nd}$ order terms to the wave kinematics or tuning the drag coefficients."*

Figures 7-16: No values are present in the x-axis. Please add labels to the x-axis of the PSDs if possible. I suggest normalizing the x-values by some physical parameter. A good choice could be the surge natural frequency.

This point is also mentioned by Referee #2. We have followed Referee #2 recommendation of eliminating the PSD units.

**Referee #2**

Referee #2

Abstract
"are showed" should be replaced by "are shown"
"My-Mz" should be removed as it depends on the frame reference that has not yet been defined
These errors have been corrected.

Figures
Frequency domains Figures are labelled "Frequency [Hz]" on the horizontal axis: the unit should be removed since values are not given for confidentiallity reasons.
We have removed the units as suggested

Conclusion
the sentence "it was used the more recent version of the SiL method" should be replaced by "we used the most recent version of the SiL method developed at CENER"

The sentence has been replaced as suggested.